# Canine Mesenchymal Stromal Cell Exosomes: State-of-the-Art Characterization, Functional Analysis and Applications in Various Diseases

**DOI:** 10.3390/vetsci11050187

**Published:** 2024-04-25

**Authors:** Evelyn Saba, Mansur Abdullah Sandhu, Alessandra Pelagalli

**Affiliations:** 1Department of Veterinary Biomedical Sciences, Faculty of Veterinary and Animal Sciences, PMAS-Arid Agriculture University, Rawalpindi 46300, Pakistan; evelyn.saba@uaar.edu.pk (E.S.); mansoorsandhu@uaar.edu.pk (M.A.S.); 2Department of Advanced Biomedical Sciences, University of Naples Federico II, Via Pansini 5, 80131 Naples, Italy; 3Institute of Biostructures and Bioimages, National Research Council, Via De Amicis 95, 80131 Naples, Italy

**Keywords:** canine mesenchymal stromal cell, exosomes, function, clinical application

## Abstract

**Simple Summary:**

Canine mesenchymal stromal cell-derived exosomes represent a potential tool for clinical and novel therapeutic approaches in both human and canine species. Characterization studies demonstrated interesting functional activity, although considerable variations in the source and culture conditions of these MSCs have been demonstrated during the production process. From this evidence and taking into account the important advantage of MSCs being less immunogenic, a new prospect is open. Future directions include a better understanding of the local and systemic roles of cell-type-specific exosomes and the mechanisms by which these exosomes are released to treat specific diseases.

**Abstract:**

Canine mesenchymal stromal cells (MSCs) possess the capacity to differentiate into a variety of cell types and secrete a wide range of bioactive molecules in the form of soluble and membrane-bound exosomes. Extracellular vesicles/exosomes are nano-sized vesicles that carry proteins, lipids, and nucleic acids and can modulate recipient cell response in various ways. The process of exosome formation is a physiological interaction between cells. With a significant increase in basic research over the last two decades, there has been a tremendous expansion in research in MSC exosomes and their potential applications in canine disease models. The characterization of exosomes has demonstrated considerable variations in terms of source, culture conditions of MSCs, and the inclusion of fetal bovine serum or platelet lysate in the cell cultures. Furthermore, the amalgamation of exosomes with various nano-materials has become a novel approach to the fabrication of nano-exosomes. The fabrication of exosomes necessitates the elimination of extrinsic proteins, thus enhancing their potential therapeutic uses in a variety of disease models, including spinal cord injury, osteoarthritis, and inflammatory bowel disease. This review summarizes current knowledge on the characteristics, biological functions, and clinical relevance of canine MSC exosomes and their potential use in human and canine research. As discussed, exosomes have the ability to control lethal vertebrate diseases by administration directly at the injury site or through specific drug delivery mechanisms.

## 1. Introduction

There is a growing interest in repairing cellular damage without the risk of adverse drug reactions or invasive surgery. For this purpose, different treatment options are being explored, including mesenchymal stromal cell (MSC) transplantation and exosome delivery, as well as gene therapy. Canine MSCs can be recovered from a variety of tissues, including bone marrow [1], adipose tissue [2,3], and umbilical cord/Wharton’s jelly [4]. Extensive research has demonstrated that MSCs have the potential to treat a wide range of conditions, illnesses, and disorders. Since exosomes have been known to be released by cells, recent research has shown that exosomes contain lipids and cytokines, as well as mRNA and microRNA [5,6,7]. Exosomes have an endosomal origin and possess a diverse proteome with a high content of tetraspanins, such as CD9, CD63, CD81, Rab5, TSG101, and flotillin [8,9]. However, certain variations in the expression of these proteins depend on the site of the cell origin [10]. Exosomes are mainly divided into small (>200 nm) and medium (<200 nm); however, some new types of exosomes have also been discovered, such as oncosomes, which are large exosomes derived from tumors [11,12]. The elucidation of the mechanisms by which exosomes interact in animal physiology and pathology may have a significant impact on the overall state of exosomes research. The production of exosomes is a conserved form of message that facilitates communication between cells in the human body, as well as between organisms of similar or distinct species [13]. 

It is important to consider why canine MSCs and their products are preferred over those of other species. The answer can be found in the previously published literature [14], which suggests that canine MSCs are used in canine therapeutics and are considered the most appropriate model for human diseases. Therefore, products obtained from canine MSCs may also be effective for human treatment. While there have been studies on feline [15] and porcine exosomes [16], the proliferation rate of canine MSCs is higher compared to that of feline and porcine MSCs when supplemented with FBS [17,18]. Hence, over the past decade, not only have the properties of canine MSC-derived exosomes been studied after cell isolation, but also their biological properties. Regarding the latter aspect, canine MSC-derived exosomes have been shown to exert anti-inflammatory effects [9] by reducing the concentration of IL-1β. Additionally, they were found to promote angiogenesis, thereby promoting cell proliferation [19] and exerting neuroprotective effects [20] in vitro and in vivo. Exosomes from serum-free cultured MSCs suppressed the pro-inflammatory cytokine production and reduced disease severity, and, in addition, antiviral microRNAs contributed to the suppression of viral replication [21]. Exosomes can be extracted from the body’s normal tissues/fluids and mimic some of the mother cell’s physiological functions and features, making them a critical tissue source for stem cell therapy [22]. This renders the source of MSC-derived exosomes a critical component of the intrinsic properties of exosomes. 

The field of isolating exosomes, not only from MSCs but also from other cells and biological fluids, is rapidly emerging in veterinary medicine. Research on exosomes is increasing year by year and various applications of exosomes as biomarkers for diagnostic and therapeutic purposes are being explored. Furthermore, human and animal physiologies are analogous and share a variety of diseases, making animals a suitable spontaneous induction model for human studies [23,24]. While the potential to use exosomes as an alternative to derived MSCs to treat various diseases opens up new opportunities, further research is needed to better understand how exosomes work and their unique biological activities. 

This review provides an overview of the biological properties of canine MSC exosomes, as well as the recent literature on their potential applications in various diseases.

## 2. Physiological Biogenesis of Mesenchymal Stromal Cell Exosomes: A Cellular Process

The International Society of Cellular Therapy in 2006 came up with the first basic criteria for identifying human MSCs i.e., plastic adherence, trilineage differentiation into osteoblasts/chondrocytes/adipocytes, and more than 95% of MSCs should be positive for CD73, CD90, and CD105 surface markers [3,25]. MSCs from different sources have different levels of differentiation and proliferation [3] due to the direct impact of the microenvironment in which they live for a long time [26]. Recently, ISCT has clarified that MSC stands for mesenchymal stromal cells [27] and, lately, a large number of studies have been published on the isolation, differentiation, and characterization of MSCs; however, there is still considerable disagreement regarding correct identification of MSCs by the ISCT criteria [28]. While a broad spectrum of positive markers has been identified to describe MSCs, there is no single marker that has been identified as being specific to MSCs. This is because the ISCT standards apply only to human MSCs and not to canine MSCs, nor those of all other species. In order to understand the function of exosomes and their potential applications in diagnosis and therapy, it is crucial to understand the process of exosome manufacturing. These cultured MSCs form lipid-bound exosomes in the extracellular compartment [29], which are key factors in different physiological and pathological processes. MSCs produce two–three types of extracellular vesicles, named exosomes, microvesicle (MVs), and apoptotic bodies (APBs). The secretory pathway begins in the endoplasmic reticulum, where proteins are produced and folded. These proteins are then transported to the Golgi apparatus, where they are organized before being sent to their final destination. 

### 2.1. Formation of Exosomes

Exosomes of different sizes in the body are produced by living cells and can be internalized by multiple organ systems during intercellular communication [30,31] with a diverse cargo of proteins, lipids, and nucleic acids. The formation of exosomes is dependent on the inward budding of the endosomal membrane, and the formation of intraluminal vesicles in the multivesicular bodies. These multivesicular bodies are further fused to the plasma membrane, resulting in the release of intraluminal vesicles/exosomes in the extracellular space [8,32,33]. 

### 2.2. Cargo Organization into Intraluminal Vessels

Cargo sorting into intraluminal vessels is a fundamental step in the exosomal biogenesis process. The Endosomal Sorting Complex Required for Transport (ESCT) machinery is responsible for the identification and binding of specific proteins and lipid substrates on the endosomal membrane. The sorting of membrane proteins into the luminal vesicles is mediated through ubiquitination, which involves the addition of a small protein, ubiquitin, to the lysine residue of target proteins. This tag is then recognized by another class of proteins called ESCTs. These endosomal sorting complexes (ESCRTs) attach to ubiquitinated cargo to ensure proper intraluminal vesicle formation and storage [34]. Four distinct ESRTs have been characterized, each of which is responsible for a distinct step in the vesicle synthesis process. Despite this, mammalian cells lacking key ESCRTs are still capable of generating intraluminal vesicles [35]. The unconventional pathways are owing to the presence of specific lipids, including lysobisphosphatidic acid, ceramides, and other lipids. It is possible that these lipids may form specialized regions of the endosomal compartment that, due to the lipid composition of the local compartment, bend inward, resulting in the formation of vesicles [36]. The in vitro studies support this hypothesis by demonstrating that the formation of vesicles in lysobisphosphatidic acid-containing liposomes depends solely on the pH gradient across the membrane.

### 2.3. Multivesicular Bodies to Plasma Membrane Fusion

Once these intraluminal vesicles are made into multivesicular bodies, the next step is to attach the multivesicular bodies to the plasma membrane, which is crucial in the exosome manufacturing process. The question then becomes, how does fusion between vesicles and plasma membrane lipid structures occur? The answer given in detail by Holz and Zimmerberg [37] is that fusion between multivesicular bodies and plasma membrane is due to interactions between SNARE proteins. Once multivesicular bodies attach to the plasma membrane, these vesicles protrude outside the cell as exosomes and are released into the extracellular space. In addition to membrane budding, there are specific local changes in plasma membrane proteins and lipids that affect membrane curvature and stiffness [38]. Exosomal release is contingent upon the activation of the cytoskeleton, but not when there is a lot of Ca2^+^ in the blood. Exosomes possess a verity of proteins that are conserved during evolution. These proteins include tetraspanins (CD63, CD81, and CD9), thermal shock proteins (HSP60, HSP70, and HSP90), tumor susceptibility gene, Alix, Clathrin, and Annexins. Serum-free culture medium maximizes the rate of exosome survival and secretion by MSCs [39].

## 3. Isolation of Canine Exosomes 

The main sources of canine mesenchymal stromal cell exosomes are adipose tissue and bone marrow. To this end, many scientists have reported methods to isolate MSCs from the above sources. Briefly, adipose tissue/bone marrow are isolated from healthy animals and cultured in vitro in the presence of FBS [9,40]. Once cells reach the desired confluence, culture supernatants are extracted for exosome isolation. It is worth noting that although the culture supernatant contains FBS, exosomal fractions obtained through any of the methods mentioned below are free of FBS. This is because the exosomes are washed several times with PBS to ensure that they are free of autologous, allogeneic, and xenogeneic components. This makes exosomes suitable for various diseases as they do not contain any harmful substances. 

In this section, we detail common methods for isolating exosomes from cell culture supernatants, with a preference for canine MSCs. It is worth mentioning here that the secretory capacity of MSCs derived from bone marrow is superior to that of canine adipose tissue. Furthermore, exosomes from bone marrow have more characterized proteins for metabolic processes than those of adipose tissue exosomes. Therefore, a preferred source of functionally rich exosomes is canine bone marrow [39]. Broadly, in canines the exosomes can be isolated from body fluids or cell cultures according to the following reported methods.

### 3.1. Ultracentrifugation

This method is considered the gold standard for isolating exosomes. The main advantage of this state-of-the-art protocol is that it generates a highly enriched exosome fraction (this includes larger vesicles which are pelleted first by a lower centrifugal force) [41]. As the name of this protocol indicates, centrifugation forces from 300× *g* to 100,000× *g* are used to obtain exosomes. Briefly, the protocol involves centrifuging any bodily fluid or cell culture medium at 300× *g* for 10 min at 4 °C. The supernatant is then discarded, and the pellet is washed with phosphate-buffered saline (PBS) by centrifugation at 10,000× *g* for 30 min at 4 °C. This is followed by another round of cold PBS washes and centrifugation at 100,000× *g* for 2 h at 4 °C. After this step, the PBS is discarded, and the pellet is resuspended in PBS and centrifuged again at 100,000× *g* for 2 h at 4 °C. After the final step of centrifugation, the PBS is discarded, and the exosome pellet is resuspended in cold PBS and stored at −80 °C until further use, as shown in Figure 1A [42]. The main disadvantages of this protocol are the maintenance of the cold chain, the availability of ultra-high-speed centrifuges, and the considerable time consumption. However, despite these shortcomings, it remains the most commonly and widely used protocol for canine MSCs [43,44].

### 3.2. Ultrafiltration

This method has also been reported for the isolation of exosomes from canine adipose tissue-derived MSCs [45] and works using a basic filtration system. The pressure gradient between the two sides of an ultrafiltration membrane causes the flow of any liquid from one side to the other. The presence of a certain degree of pressure facilitates the passage of water and small, dense molecules through the membrane, thereby preventing the material from entering the pores and forming a concentrated solution for purifying, separating, and isolating the exosomes, as illustrated in Figure 1B. This is called ultrafiltration. The reliability of this method is determined by the specific membrane pore size. Generally, large diameter particulates (from 0.45 to 0.8 µg) are filtered initially, producing a small diametrical exosome-rich filtrate. Filtering with pore sizes ranging from 0.1 to 0.22 µm can be employed to obtain higher fractions [46]. This method can be employed on its own or it can be employed in conjunction with ultrasound-based filtration. In such a case, upon completion of the final centrifuge step, the pellet can be resuspended in PBS and filtered through an ultrafiltration membrane to differentiate between small- and large-sized EVs. The main issues with this method are the small sample size, co-presence of proteins and RNA with the exosomes, poor quality exosomes, and low yield [47,48,49].

### 3.3. Polymer Precipitation 

This process utilizes the general precipitation principle, in which solvents are used to modify the polarity of exosomal constituents and their solubility, causing these components to precipitate into solution, as illustrated in Figure 1C. Acetate, polyethylene glycol (PEG), protamine, protein organic solvent precipitation, and a variety of other reagents are widely used for precipitation. PEG is one of the most widely used reagents. PEG promotes exosome precipitation by reducing the solubility of exosomes. After precipitation, low-speed centrifugation is used to obtain exosomes. This method is highly stable and produces high-quality exosomes [50]. Nonetheless, the procedure does not eliminate the possibility of exosomal contamination by lipoprotein or another microparticles. Precipitation of exosomes is achieved using commercially available kits that follow the same principles as described above [51]. 

### 3.4. Size-Exclusion Chromatography (SEC)

This method of separation is based on the principle of molecular weight separation. In this approach, a mobile phase exosomal sample is added to one of the edges of a chromatography column containing porous beads, e.g., Sepharose, Sephadex, Sephacryl, or BioGel-P, which is considered as a stationary phase. High-molecular-weight particles are not able to penetrate the pores of the gel, resulting in a faster elution rate. Conversely, low molecular weight particles are able to pass through the pores more easily, resulting in slower elution rates, as explained in Figure 1D, and the adsorption of the mobile phase to the stationary phase, which is more fragile than the sample components. The dynamics of the SEC are determined by the size and morphology of the exosome isolate. If the isolate is not perfectly spherical, the elution steps of the chromatogram may be affected. The fractionation mechanism flow rate also influences the separation efficiency. The lower linear velocities facilitate the isolation, resulting in improved integrity, specificity, and function of exosomes [52]. The primary benefits of this technology are the elimination of contaminants from the exosomes, the time effectiveness, and the simultaneous processing of multiple samples. However, one disadvantage of this technology is that it is unable to distinguish between microvesicles from the same-size exosomes [53]. Therefore, it can be used in conjunction with immuno-affinity technology for the isolation of exosome subtypes.

### 3.5. Immunomagnetic Bead-Based Method 

One of the most widely used exosome capture methods based on immunoaffinity is immunomagnetic beads. The process of creating immunomagnetic beads starts with the addition of antibodies on the surface of the magnetic beads. This allows the antigen and antibody response to select specific exosomes for capture. There are several protein markers on the exosomes, including CD63 antigen, CD9 antigen, and CD81 antigen, which differentiate them from other exosomes (EVs). Therefore, immunomagnetic beads that are coated with specific antibodies bind to specific exosomes by recognizing these protein markers. Once the targeted exosomes have been captured, they are then eluted and collected [54]. The protocol for exosome isolation with immunomagnetic beads, as outlined in Figure 1E, includes the following steps: (1) preparation of the beads; (2) binding of the exosomes to the beads; (3) removal of any impurities from the exosomes, as well as the immunomagnetic beading complex; and (4) exosome elution. This protocol enables the isolation of specific subpopulations of exosomes, resulting in high purity, specificity, and structural integrity. Despite its advantages, this protocol also has some disadvantages, such as being time consuming, having an unpredictable multi-step workflow, and requiring manual handling [55]. 

### 3.6. Microfluidic-Based Method

Microfluidics is the process of controlling small volumes of fluid using micro- and nano-fabricated channel structures. This method is based on the principles of size, density, and immunoaffinity of exosomes isolated by microfluidic equipment. There are two distinct types of exosome separation methods: immunoaffinity-based isolation and a combination of microfluids with dielectrophoresis and acoustic waves, as illustrated in Figure 1F.

#### 3.6.1. Immunoaffinity-Based Isolation

This technique uses immobilized antibodies on a microfluidic surface to pre-enrich exosomes in blood samples or cell cultures [56,57]. For example, Kanwar et al. [58] developed a device, called ExoChip, which is capable of specifically isolating CD63-specific exosomes from serum within 1 h. The device comprises elongated channels connected to circular capture chambers, which extend the retention period of exosomes within the channel to maximize interaction with antibodies on the surface of the chip. The surface of the chip is coated with antibodies against CD63, an exosome marker protein. This method allows for simultaneous exosome isolation, quantification, and characterization.

#### 3.6.2. Microfluidics with Acoustic Fields and Electrophoresis

The unique feature of acoustic waves is that they are highly biocompatible and can be accurately controlled [59,60]. Therefore, acoustic waves are the best method for size-based component/particle separation. Standing acoustic waves are generated with the help of interdigitated microelectrodes, which convert electrical signals into mechanical stresses that are transmitted along the surface of the piezoelectric substrate material. The standing acoustic waves inside the microfluidic channel can create a series of pressure nodes. The basic principle of acoustic-based separation is that the particles that will flow through the channel with be encountered by the pressure nodes. This force guides the particles slightly away from the center of the channel. The displacement distance is dependent on the particle size and density. Henceforth, particles of different sizes will be transferred from various sources, resulting in particle separation [61,62]. Wu et al. [63] demonstrated a separation platform that combines surface acoustics with microfluidic components for isolating exosomes from whole blood samples without labels or contacts. The method consists of two parts. The first part involves the extraction of cells to enrich the exosomes. The second part is the extraction of apoptotic bodies and microvesicles from the exosomes. In this method, the exosomes are exposed to sound waves for a limited period of time. However, there is a slight drawback to this approach: because this phenomenon operates according to the principle of acoustic impedance, other cellular constituents with similar acoustic impedance to exosomes will have a direct impact on the isolation [63]. 

#### 3.6.3. Electrophoresis

Electrophoresis is the process of collecting exosomes through a nano-membrane. In general, an electrophoretic field is typically applied to a dialysis membrane of approximately 30 nm. In the presence of an electric field, particles such as proteins and other molecules will permeate the membrane, while exosomes will be trapped within the membrane and separated [64]. This method offers a range of benefits, including a low chemical and reagent consumption, high exosomal purity, rapid separation, and a high detection rate [65]. However, it also has some drawbacks, such as the need for specialized equipment, in addition to the use of microfluidic chips [66].

## 4. Characterization of Exosomes

Since exosomal characterization is a discrete process, several parameters can be used to determine whether isolated particles are exosomes. According to the International Society for Extracellular Vesicles, two types of protein on exosomes are indicative of their true presence. Exosomes are composed of a plasma membrane containing transmembrane proteins, also known as GPI-anchored proteins and cytosolic proteins [12]. It is necessary to determine the purity of exosomes obtained from these biological fluids by distinguishing the presence and absence of various non-structural protein components. In general, exosome characterization is carried out on three levels: exosome morphology, exosome size, and exosomal surface protein marker. Exosome characterization methods are divided into two categories: external characterization (particle size and morphology) and internal characterization (presence of membrane proteins and lipid morphology) [67].

In the following section, we describe common exosome characterization approaches using external or biophysical methods, as well as internal or protein detection approaches, specifically for canine MSC exosomes. Notably, TEM and Western blot analysis are the most commonly used methods for characterization of canine adipose tissue-derived MSC exosomes, as reported by Merlo et al. [68].

### 4.1. Transmission Electron Microscopy (TEM)

This method is employed to comprehend and evaluate the morphology of isolated exosomes. It is also used to detect the presence of contaminants. The most common method for visualization of exosomes by TEM is the negative staining procedure (uranyl acetate), as illustrated in Figure 2A. However, during preparation, drying results in the typical collapsed vesicle or cup-shaped morphology [69]. To address this issue, CryoTEM is the gold-standard method for imaging biological objects. This technique rapidly freezes precious cellular structures through hydration, thus providing a more detailed visualization of EVs from non-vesicular particulates. More advanced techniques include combining TEM with immunogold labeling to aid in the identification of exosomes and EVs. Additionally, the use of atomic force microscopy and microarray technology can yield comprehensive information on the size and structural characteristics of EVs [70,71]. An et al. and Villatoro et al. have used this technique to characterize exosomes from canine adipose tissue and colostrum [44,72].

### 4.2. Atomic Force Microscopy (AFM)

This method quantifies the size of exosomes (or nanoparticles) through the interaction between a probing tip and a sample surface, as illustrated in Figure 2B. This method is a viable alternative to electron diffraction for exosomal characterization. The advantage of this method is that it can accurately measure the size of exosome without altering its natural state. This method has been widely used for the quantitative examination of exosomes derived from saliva [73], blood [74], and synovial fluid [75]. 

### 4.3. Nanoparticle Tracking Analysis (NTA) 

This method is frequently employed to assess the physicochemical properties of exosomes (shape, size, density, porosity, and surface charge), which are strictly related to their biological interactions and suitability for therapeutic applications. Biophysical techniques were employed to characterize the size range of exosomes. The NTA method was used to measure the concentration and size distribution of exosomes. Exosomal particle velocity is determined by exosomal movement, which is the Brownian motion of nanoparticles suspended in a liquid base [76]. This movement can then be compared to the particle size. This method allows the measurement of the exosomal particle through image analysis, as illustrated in Figure 2C. The larger the particle, the more slowly it will move, and vice versa. Henceforth, this method enables the measurement of EV sizes with diameters as small as 30 nm. This method is particularly appealing due to its simple sample preparation and recovery of exosomes in their original form after measurement. Additionally, this method can be employed to detect the presence of antigens by the use of fluorescently labeled antibodies [77]. This method is most commonly used to characterize canine MSC exosomes, as supported by the previously reported literature [40,78].

### 4.4. Dynamic Light Scattering (DLS)

This method, also referred to as photon correlation spectroscopy, is an alternative method for determining exosomal size. It involves passing a monochromatic laser beam through a suspension of particles, resulting in fluctuations in scattering intensity over time due to Brownian motions of the particles within the sample, as exemplified in Figure 2D. This method is advantageous because it is capable of measuring particle sizes as small as 1 nm, making it particularly suitable for single-cell exosome measurements, such as those of RBCs [79]. This method permits the determination of vesicle diameters, but it does not provide quantitative information on the biochemical composition of exosomes. 

### 4.5. Tunable Resistive Pulse Sensing (TRPS)

This technique has recently been developed and is slightly different from its counterparts. This technique is based on the principle of resistive pulse sensing, measuring the current flow through a small aperture conjugated with tunable nanopores, which allows the passage of ionic current with the particles under question by adjusting the pore size, as shown in Figure 2E [80]. This method quantifies the size and abundance of exosomes, ranging from approximately 50 nm in diameter to cell size. Consequently, this method is useful when investigating the entire cell function and uptake. The primary advantage of this method is that it enables the in situ measurement of individual particulate exosomes [81]. Like all other methods, this method has some limitations, such as sensitivity to system stability, where particles can easily clog pores, and other sensitivity issues, such as the effect of small particle size on high background noise in the system [82]. This approach has been widely used to identify leukemic exosomes bound to the extracellular matrix [83].

### 4.6. Flow Cytometry 

Flow cytometry typically uses light scattering and fluorescence measurements to qualitatively analyze individual particles in a sample. In this method, beads with antigens attached to their exosomal membranes are conjugated to antibodies. The beads are further conjugated to a fluorophore-conjugated secondary antibody, and the resulting combination of beads and exosomes is suspended in a liquid medium. When a laser beam excites a fluorophore (as illustrated in Figure 2F), the fluorophore fluoresces at longer wavelengths. Exosomes are small in size and can be analyzed by flow cytometry. Therefore, flow cytometry must meet certain criteria to be effective and reproducible [84]. 

Antibody-based methods are used to identify the intrinsic characteristics of exosomes and are as follows:

#### 4.6.1. Western Blot

Exosomes are protein-containing vesicles. Therefore, Western blotting can be used to check the presence and identification of exosomes, as shown in Figure 2G. This technique can give information on the yield and purity of exosome preparations. Additionally, the molecular weight of the target protein can also be determined. However, to ensure adequate sensitivity and control for antibody specificity, Western blotting requires large sample volumes [85]. Canine MSCs were characterized by Western blot, showing exosomal protein markers using a mouse anti-TSG101 monoclonal antibody that cross-reacts with canine TSG101. Anti-TSG-6 antibodies were also used to characterize exosomes from cMSCs [39,44].

#### 4.6.2. ELISA

The Sandwich ELISA is a well-established method for the detection of antibodies based on a multi-well format, as illustrated in Figure 2H. It is most commonly used for exosomal protein detection [44,86]. The dissociation-enhanced lanthanide fluorescence immunoassay is a time-resolved fluorescence assay that uses a single antibody assay to detect exosome-associated molecules [87]. Surface plasmon resonance [88] and interference imaging [89] are other methods used for exosomal protein characterization, but require specialized equipment, which limits their use and scope.

## 5. Characterization of Canine MSC-Derived EVs and Possible Factors Influencing Their Biological Properties

Canine exosomes (often in the form of EVs), can be obtained from several sources (bone marrow, adipose, amniotic and gingival tissues, umbilical cord), and their physical and functional characteristics have been studied. These products exhibit striking and unique characteristics, such as the expression of ALIX and TSG101, which are thought to be involved in exosome biogenesis process for ESCRT-facilitated transport (Table 1) [39]. Furthermore, taking into account the source of MSCs, exosomes obtained from bone marrow were 13-fold higher than AD-MSCs, which was also confirmed by proteomic analysis. Considering that cBM-MSCs and cAD-MSCs are almost equivalent in their ability to inhibit T cell activation, this observed difference does not appear to reflect a specific biological property [90,91]. Studies focused on increasing the number of exosomes collected have shown that EVs are more abundant in P0 of cultured amniotic MSCs compared with later passages, suggesting that this passage is most suitable for application in pre-clinical testing [92]. Notably, the same exosomes isolated from amniotic MSCs, if used as a supplement to MSC cultures, could increase the cell expansion rate because P1 cells have more metabolic activity compared to P2 cells [93].

Regarding the biological potential of canine EVs isolated from amniotic fluid MSCs, procoagulant activity was observed in their parental cells and human MSC-EVs [94]. Stronger EV activity was observed at early passages compared with later passages, although the true cause of this biological activity is not fully understood. Furthermore, studies of the properties of EV under hypoxic conditions have shown that EVs increase COX-2 content, thereby ameliorating inflammation through paracrine effects on canine macrophage cell lines [95]. 

It is worth nothing that research interests for potential therapeutic modalities using MSC-EVs have been identified in canine species. To improve the quality of secretomes for therapeutic use, liquid secretomes were produced using canine adipose MSCs under the supervision and authorization of Istituto Zooprofilattico Sperimentale Lombardia and Emilia-Romagna, IZLER, Brescia, Italy. This product has been shown to be safe for potential use in the treatment of canine osteoarthritis and does not cause local or systemic adverse effects [96].

**Table 1 vetsci-11-00187-t001:** Main properties of canine mesenchymal stromal cell-derived exosomes.

Source of Canine MSCs	Characteristics and Specific Properties of Exosome/EVs Isolated	Reference
Bone marrow	– Comparative quantification of exosomes collected from MSCs demonstrate a higher percentage for BM respect to Ad-MSC;	[39]
	– Exosome release can be controlled by using cationized gelatin hydrogels;	[43]
	– Exosomes can reduce proinflammatory response	[43]
Adipose tissue	– EVs exert immunosuppressive effects on stimulated CD4+ T cells in vitro	[45]
Umbilical cord	– The number of EVs isolated is not influenced by the conditioned medium (CM) storage;	[97]
	– EVs isolated from conditioned medium storage (−80 °C) show a morphology similar to that immediately isolated	[97]
Amniotic tissue	– Exosomes show antiapoptotic properties with the transport and transfer of telomeric DNA;	[92]
	– Exosome secretion is negatively influenced by cell age, injury exposure to inhospitable microenvironment	[92]
Gingival tissue	– Exosomes characterized by TEM, nanoflow analysis, and Western blotting show typical characteristics;	[98]
	– Their role has been demonstrated in the promotion of both proliferation and migration of Madin–Darby canine kidney cells	[98]

## 6. Therapeutic Possibilities/Potential of EVs in Multiple Diseases

### 6.1. Application of Exosome in Bone Disorders

Injuries to the bone caused by trauma or disease are typically accompanied by soft tissue injury. The process of bone healing is complex and meticulous, involving the creation of a new bone through a variety of cellular and molecular pathways. The development of the vasculature is important for the growth and regeneration of the bones. It is responsible for hormones, growth hormones, oxygen, nutrients, and metabolites, as well as the transportation of these components. MSC exosomes help to create the angiogenesis in the area of the bone that has a defect [99], which is important for bone regeneration. Exosomes have been demonstrated to have therapeutic potential in canine bone disorders, and are associated with bone and cartilage formation, metabolism, and pathological alteration [100]. Furthermore, they can be used as alternatives to conventional treatments and biomarkers for diagnosis [101]. Modified exosomes have demonstrated robust bone-targeting abilities, increased efficacy, and avoidance of systemic adverse effects [102]. Furthermore, exosomes possess excellent biocompatibility, biofilm penetration, and therapeutic properties [103]. They can be used as drug delivery vehicles targeting the bone microenvironment [104]. Exosomes have also been studied in bone tissue engineering, with the potential to be used as a cell-less therapy in combination with tissue-engineered bone [105]. MSC exosomes activate a variety of signaling pathways, including PI3K/AKT, Wnt/β-catenin, and BMP/Smad, which promote bone remodeling by stimulating osteoblast proliferation [106]. A study by Liang et al. [93] shows that bone defects can be repaired by increasing the amount of angiogenesis in the area and preventing bone resorption by activating the AKT/mTOR signaling pathway. Moreover, macrophages and other non-stem cells release greater amounts of BMP-2 and other growth factors when exosomes from MSCs were added [107]. Exosomes excreted by cells in the bone microenvironment play a role in the maintenance and regeneration of bone homeostasis (Figure 3A) [108]. Furthermore, biomaterials may be employed as carrier agents for the delivery of exosomes to bone defect sites.

Exosomes are composed of a range of molecules, including proteins, lipids, and microRNAs, that can interact with bone cells and influence bone development. For example, numerous studies have demonstrated that exosomes from bone marrow MSCs can enhance vascular function, blood circulation, and bone remodeling. Expression of HIF-1a in MSCs induces exosome secretion, leading to expression of genes related to osteogenesis, and tRNA-10277 regulates adipogenesis and osteogenic potential [109,110]. For example, miR-935-enriched exosomes promote osteoblast growth by regulating STAT1 signaling and transcriptional activation [111], while miR-1260b inhibits osteoclast activity by targeting the Wnt5a-mediated RANKL pathway [112].

Osteoclasts are also involved in bone resorption by activating the production of exosomes (containing miR-214). These exosomes are identified by the Ephrin receptor (Ephrin A2/Ephrin A2 receptor 2) and enter the osteoblasts, inhibiting EphA2 activity and promoting the resorption process [113]. Osteoclast exosomes have been shown to transport signaling molecules, including RANKL, into osteoblasts. This, in turn, induces activation of NF-κB and other osteoclastogenetic pathways, thereby enhancing osteoclastogenicity and bone remodeling [114] by TRPV4 receptor inhibition [115].

### 6.2. Exosomes Use in Various Pathological Conditions of Canine Skin

The skin is a large organ of the body that acts as a barrier between the internal and external environment. It consists of three distinct layers: the epidermis, the dermis, and the hypodermis, each of which is composed of a variety of cell types. Systemic administration of allogeneic MSCs has been proposed as a potential treatment for reducing canine atopic dermatitis [116]. Exosomes have been investigated for their potential use in treating various pathological conditions in canine skin, including atopic dermatitis, wounds, and aging [117]. In particular, recent literature suggests improved research on the use of exosomes for various skin injuries repair [118], and new data suggest that exosomes play a pivotal role in promoting wound healing by increasing neovascularization in a large canine laceration model. The potential repair mechanism may be related to several mechanisms: 1. angiogenesis in the affected area [119]; 2. regulation of the hydration status of the stratum corneum [78]; and 3. reduction in skin inflammation through immunosuppression [12]. To enhance the long-term use of cMSC exosomes in isolation, a cryoprotectant (sodium carboxymethylcellulose) was used to prepare exosome gels and treat skin wounds. The experimental results of this study demonstrated that the exosome gel promoted wound healing with scar-free and organized collagen, thus suggesting its potential use in treating cutaneous wounds [120]. 

### 6.3. Exosomes Treatment in Canine Reproduction

Artificial insemination is a widely used method in large mammals; however, due to the unique reproductive physiology of canines, there are limitations to its use. Dogs are the only mammals whose ovarian follicles secrete immature oocytes during prophase-I. These immature oocytes take 48–72 h to reach the fallopian tube maturation stage. This is why the environment surrounding the immature oocyte is of great importance for its maturation. A study concluded that the extracellular vesicles containing miRNAs [121] play an essential role in the maturation of canine oocytes in the oviduct. The cryopreservation of male dog sperm can lead to a change in the morphology of the sperm, resulting in a decrease in the post-thaw fertilization capabilities. Qamar’s hypothesis suggests that canine MSC supplementation may be able to repair the damage caused by the freezing process [122], resulting in improved mobility and viability.

At present, there is limited research on the effects of exosomes on endometritis in female dogs; however, in humans, exosomes are thought to be able to treat endometritis, potentially providing novel diagnostic and therapeutic strategies. When exosomes are blocked, the functions of promoting neural and angiogenic development are impaired [123], as the development of nerves and vessels is a critical step in the development of the endometrium [124].

### 6.4. Exosomes Use in Canine Digestive Problems

Like that of other species, dog colostrum contains immunoglobulins, proteins, or fats, as well as various biomolecular structures (exosomes) associated with signaling pathways and cell-mediated communication with neonatal tissues [125]. Exosomes are able to withstand the digestive tract enzymes of a newborn, reach the intestinal cells, and promote the intestinal cell proliferation, thus stimulating the development of the newborn’s gastrointestinal tract [126]. In a subsequent study of canine colostrum in vitro, Villatoro and colleagues observed an association between the mother’s exosomes and the intestinal MSCs of the pup. Furthermore, exosomes presented an antioxidant capacity on the fibroblasts in response to reactive oxygen species activity in the cell, suggesting that exosomes play an essential role in the growth and development of dogs during the early stages of life [72]. In a clinical trial, pretreatment of canine adipose MSCs with TNF-α resulted in increased release of immunomodulatory factors such as TSG-6 and PGE2, which have been implicated in inducing phenotypic changes in macrophages [127]. This type of treatment may be beneficial in treating irritable bowel syndrome (IBS) in humans.

### 6.5. Exosomes Use in Canine Cardiovascular System Problems

Exosomes play a critical role in the regulation of disease progression by transporting and exchanging signaling molecules [128]. Fibroblasts in the heart secrete exosomes, which provide protection from the effects of ischemia and perfusion injury, including the prevention of pyroptosis [129] and apoptosis.

### 6.6. Exosomes as Emerging Diagnostic Biomarkers

The cancer research approaches continue to search for new and early biomarkers of the disease to develop and organize the best strategies to combat it. In this regard, Brady et al. [130] demonstrated, on the basis of proteomic analysis, that serum exosomes can be used as precocious biomarkers of canine osteosarcoma. In fact, the proteomic cargo of serum exosomes showed differences between healthy dogs and dogs with osteosarcoma. Similar analysis was performed on serum exosomes isolated from dogs affected by Leishmaniosis and associated alterations demonstrating that the decrease in exosomal miR-122-5p, associated with serum levels of high-density lipoproteins, and increased serum levels of low-density lipoproteins could be a predictive indicator of disease [131]. Notably, a recent paper demonstrated for the first time that miRNAs can be used as tumor diagnostic tool for cross-species differentiation, classifying human and canine lymphoid tumor cell lines [132]. Moreover, this diagnostic approach has been shown to be helpful in excluding pulmonary metastasis. More recently, a research study by Ju-Hyun et al. [133] focused on evaluating the reduction in the inflammatory pathway of the well-known inflammatory bowel disease (IBD), and showed the efficacy of EVs secreted from canine adipose MSCs. In particular, EVs depleting TSG-6, considered to be a major factor in regulating inflammatory responses, reduced the inflammatory conditions associated with DSS-induced colitis in mice by improving the Treg population in the inflamed colon. 

As summarized in Figure 3B, canine exosomes showed potential therapeutic activity against a variety of diseases. Exosomes not only act as bilipid membrane particles, but also as carriers, transporting various substances to specific targets, thereby promoting their enhanced beneficial abilities.

## 7. What Are the Advantages and Disadvantages of Using Mesenchymal Stromal Cells or their Exosomes?

Mesenchymal stromal cells (MSCs) are a diverse group of cells that are capable of self-differentiation into mesodermal lineage tissues. The regenerative potential of MSCs after in vivo transplantation into animal models has led to speculation that they may have the therapeutic potential to treat human diseases. MSCs are capable of secreting soluble immunomodulatory factors via paracrine pathways and inhibiting host vascular responses via contact-dependent regulation [134,135], thus altering the balance of alloreactivity from effector to regulatory functions. The growing experimental and clinical evidence suggests that exosomes may be the next generation cell-free therapeutics, offering attractive benefits over MSCs in terms of non-tumor formation, immunogenicity, and stimulation of angiogenesis. The use of MSC and their formed exosomes has distinct benefits and drawbacks, which are outlined in Table 2.

Stem cell-based therapies should be used with caution and all potential adverse reactions should be taken into account. Stem cell transplant patients receive long-term chemotherapy or radiation to suppress their immune system and reduce tumor chances [145]. Exosomes have been used to diagnose a variety of diseases, and their application in disease treatment, especially in veterinary medicine, has been widely discussed by Kandeel M. et al. and Heidarpour et al. [146,147]. Canine MSC-derived exosomes have emerged as promising therapeutic agents, but their use also poses different challenges. To fully realize their potential, we need to standardize and validate the most appropriate isolation methods and perform comprehensive functional characterization to determine their potential as diagnostic biomarkers. Furthermore, it is important to conduct rigorous safety and consequential studies to ensure their safety and effectiveness. We believe that by directly tackling these issues, we can maximize the benefits of dog exosomes and offer a safe, effective treatment option.

## 8. Conclusions

Compared with direct administration of MSCs, exosomes have considerable therapeutic potential to address a variety of pathological conditions. Exosomes may be less immunologically relevant to pathologic conditions in dogs and humans, so allogeneic exosomes could be used in other individuals and species. Future studies need to elucidate the specific mechanisms and effects of exosomes at the tissue level. This is important for the improvement in their functionalization with specific relevance, although the current literature only considers them as new strategic tools for clinical applications rather than the MSCs from which they originate.

## Figures and Tables

**Figure 1 vetsci-11-00187-f001:**
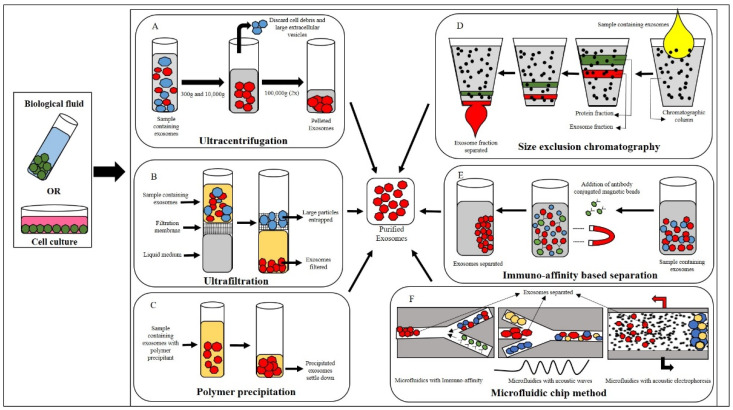
(**A**–**F**) Summary of methods for isolating canine MSC exosomes from biological fluids or cell cultures.

**Figure 2 vetsci-11-00187-f002:**
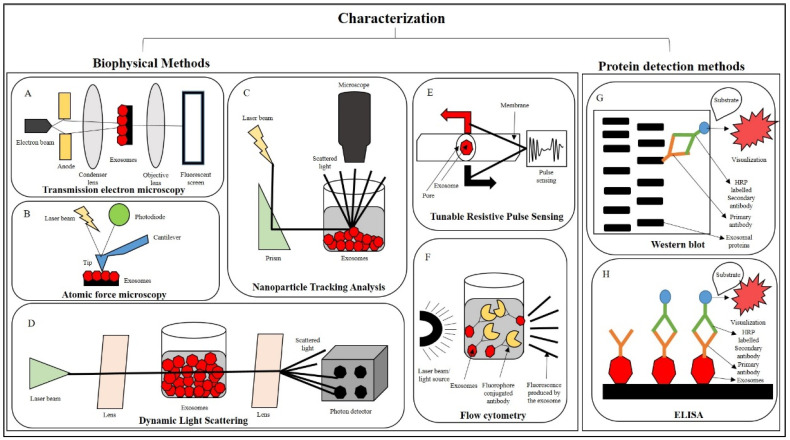
(**A**–**H**) Exosomal characterization by external or biophysical methods: a brief overview.

**Figure 3 vetsci-11-00187-f003:**
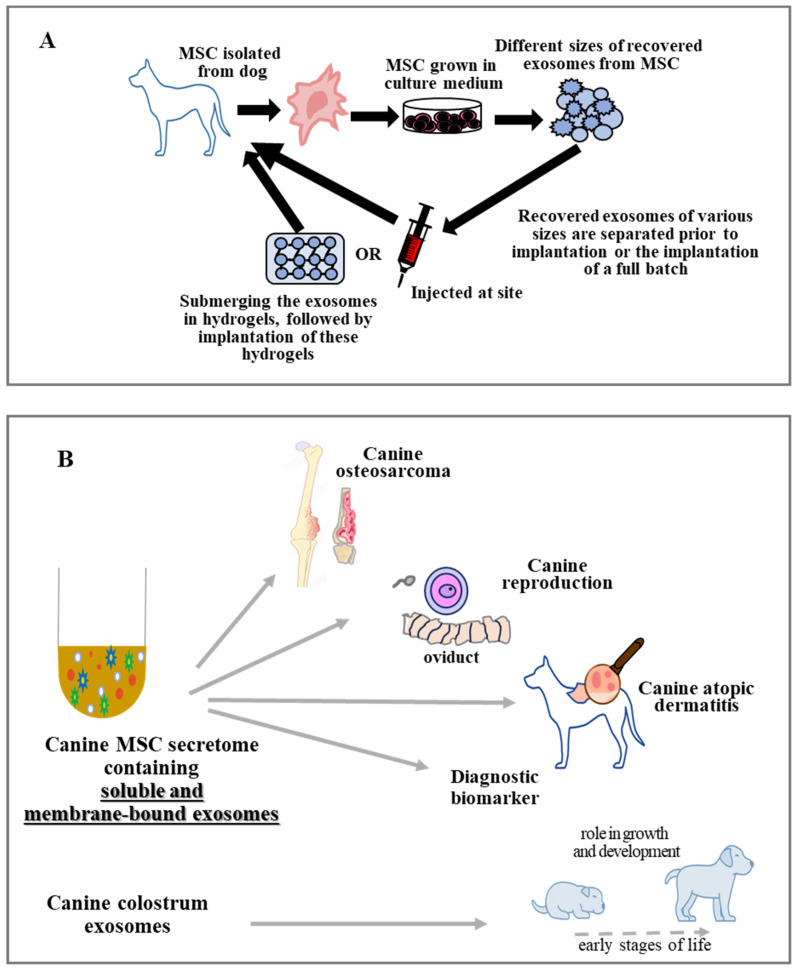
(**A**,**B**) Isolation of canine exosomes and their potential application in the treatment of diseases.

**Table 2 vetsci-11-00187-t002:** Advantages and disadvantages of the isolation and use of mesenchymal stromal cells and their exosomes in research and clinical applications.

	Advantages	Disadvantages
**Mesenchymal stromal cells (MSCs)**	° Isolated from freshly sacrificed food animals and from multiple organs [25,136,137]; ° Rapidly proliferative and easily transplanted; ° Immunomodulatory properties; ° Fear of viruses and prion presence due to the use of pooled FBS/platelet lysate in culture medium; ° Less ethical issue of their use in research [137].	° MSC transplantation involves tumor progression due to production of angiogenic factors [138]; ° Risk of morphology change decrease and change in telomerase activity (after long-term culture) [139]; ° Short-lived viability for transplantation after intravenous injection [140].
**Exosomes**	° Suppress different immune related cells and promote tissue repair [43]; ° Low risk of aneuploidy [43]; ° Relatively no ethical issues stated so far [43]; ° Ready to use off-shelf availability and transplanted MSCs do not exhibit any unpredictable behavior [140]; ° No evidence of impaired cell survival or a decrease in cell number [141]; ° Optimal tool to diagnose certain conditions [128,142].	° No established protocol for the isolation of exosomes [128]; ° Research is requested to treat different illnesses [128]; ° Natural exosomes are not very good at delivering drugs (low stability and easy to break down) [128]; ° Technical problems in their preparation (clogged filters and reduced life of the membranes) [143]; ° No high quantity of produced exosomes and difficulty in obtain exosomes of different sizes [144].

## Data Availability

Not applicable.

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
