# Peer review of "Canine Mesenchymal Stromal Cell Exosomes: State-of-the-Art Characterization, Functional Analysis and Applications in Various Diseases"

_vetsci, 2024, doi:10.3390/vetsci11050187_

Round 1

Reviewer 1 Report

Comments and Suggestions for Authors

Review titled: Canine Mesenchymal Stem Cell Exosomes: State-of-the-art characterization, functional analysis and applications in various diseases.

In the manuscript as a whole, but more specifically in item 2: Physiological biogenesis of mesenchymal stem cell exosomes: a cellular process, there needs to be more clarity regarding what mesenchymal stem cells and mesenchymal stromal cells are. Practically all published studies with exosomes are obtained from mesenchymal stromal cells from different tissues. The position statement on the nomenclature of the committee from The International Society for Cell & Gene Therapy (ISCT) can be of great value in further clarifying this definition (Viswanathan S, 2019). After the authors have well-substantiated this concept, it would be essential to check whether the title of the manuscript agrees with the nomenclature and all the citations of mesenchymal stem cells and mesenchymal stromal cells throughout the text.

  2.3 Multivesicular bodies to Plasma Membrane Fusion: “Briefly, adipose tissue/bone marrow was isolated from healthy animals 151 and cultured in vitro in the presence of FBS [9,34].” Many published works use FBS only at the beginning of cell expansion. The day before obtaining the conditioned medium containing the exosomes, washes are carried out with PBS, and the FBS is removed from the medium. This procedure could be better detailed because the presence of FBS in the medium can generate a bias in the analyses.

Finally, since the title is focused on exosomes derived from canine MSCs, the manuscript could be concluded by discussing more about the specific challenges and real perspectives for using this technology in veterinary medicine.

References:

Viswanathan S, Shi Y, Galipeau J, Krampera M, Leblanc K, Martin I, Nolta J, Phinney DG, Sensebe L. Mesenchymal stem versus stromal cells: International Society for Cell & Gene Therapy (ISCT®) Mesenchymal Stromal Cell committee position statement on nomenclature. Cytotherapy. 2019 Oct;21(10):1019-1024. doi: 10.1016/j.jcyt.2019.08.002. Epub 2019 Sep 13. PMID: 31526643.

Author Response

Reviewer# 1:

In the manuscript as a whole, but more specifically in item

1: Physiological biogenesis of mesenchymal stem cell exosomes: a cellular process, there needs to be more clarity regarding what mesenchymal stem cells and mesenchymal stromal cells are. Practically all published studies with exosomes are obtained from mesenchymal stromal cells from different tissues. The position statement on the nomenclature of the committee from The International Society for Cell & Gene Therapy (ISCT) can be of great value in further clarifying this definition (Viswanathan S, 2019). After the authors have well-substantiated this concept, it would be essential to check whether the title of the manuscript agrees with the nomenclature and all the citations of mesenchymal stem cells and mesenchymal stromal cells throughout the text.

Author response: Respected reviewer, thank you for your comment. After detailed reading on the clarification study by Viswanathan S, 2019, we have added the description of MSCs as Mesenchymal Stromal Cells on pages 2 & 3, (lines 96-98). Subsequently we have homogenised the clarified term in the title and throughout the manuscript file.

  1. Multivesicular bodies to Plasma Membrane Fusion: “Briefly, adipose tissue/bone marrow was isolated from healthy animals 151 and cultured in vitro in the presence of FBS [9,34].” Many published works use FBS only at the beginning of cell expansion. The day before obtaining the conditioned medium containing the exosomes, washes are carried out with PBS, and the FBS is removed from the medium. This procedure could be better detailed because the presence of FBS in the medium can generate a bias in the analyses.

Author response: Respected reviewer, thank you for your comment. We have given the detailed description about removal of FBS from exosomal fractions on page 4, (lines 161-165) respectively.

  1. Finally, since the title is focused on exosomes derived from canine MSCs, the manuscript could be concluded by discussing more about the specific challenges and real perspectives for using this technology in veterinary medicine.

Author response: Respected reviewer, thank you for your comment. We have given the reference for canine MSCs application with respect to veterinary medicine [147-148] and furthermore we have added the specific challenges for this technology on page 16, (lines 614-623) respectively.

Reviewer 2 Report

Comments and Suggestions for Authors

This is a review about canine Mesenchymal Stem Cell Exosomes and their applicability in different diseases. It is a well organized and structured review that is important to the field and may be useful for a better understanding before its application in a clinical setting. 

For me, it as manuscript that has all the potential to be published as a review.

Author Response

Reviewer# 2

This is a review about canine Mesenchymal Stem Cell Exosomes and their applicability in different diseases. It is a well-organized and structured review that is important to the field and may be useful for a better understanding before its application in a clinical setting.

For me, it as manuscript that has all the potential to be published as a review.

 Author response: We thank the reviewer #2 for the time spent for the revision of our manuscript and for the appreciation of the quality and the well organization of the manuscript. 

Reviewer 3 Report

Comments and Suggestions for Authors

A manuscript (ID: vetsci-2933655) enitled: “Canine Mesenchymal Stem Cell Exosomes: State-of-the-art characterization, functional analysis and applications in various diseases has been submitted by the authors Mansur Abdullah Sandhu, Evelyn Saba and Alessandra Pelagalli to the journal Veterinary Sciences (Veterinary Physiology, Pharmacology, and Toxicology).

The draft documents a solid piece of scientific work and should fit in a perfect way to the topics of the chosen target journal. However, I still have some remarks concerning the literature discussed. It is not quite clear to me why the authors have ignored an older publication regarding this topic. It would be of interest to learn a bit more of the progress in the field during the last ten years.

Evelien de Bakker, Bernadette Van Ryssen, Catharina De Schauwer & Evelyne Meyer (2013) Canine mesenchymal stem cells: state of the art, perspectives as therapy for dogs and as a model for man, Veterinary Quarterly, 33:4, 225-233, DOI: 10.1080/01652176.2013.873963

The authors focus on canine mesenchymal stem cells which are promising objects for the therapeutic approaches under study. Nevertheless, I have missed a detailed explanation what are the advantages of canine mesenchymal stem cells in comparison to stem cells from porcine or feline origin.     

Zhao, Q.; Zhang, X.; Li, Y.; He, Z.; Qin, K.; Buhl, E.M.; Mert, Ü.; Horst, K.; Hildebrand, F.; Balmayor, E.R.; et al. Porcine Mandibular Bone Marrow-Derived Mesenchymal Stem Cell (BMSC)-Derived Extracellular Vesicles Can Promote the Osteogenic Differentiation Capacity of Porcine Tibial-Derived BMSCs. Pharmaceutics 2024, 16, 279. https://doi.org/10.3390/pharmaceutics16020279

Sung SE, Seo MS, Kang KK, Choi JH, Lee S, Sung M, Kim K, Lee GW, Lim JH, Yang SY, Yim SG, Kim SK, Park S, Kwon YS, Yun S. Mesenchymal Stem Cell Exosomes Derived from Feline Adipose Tissue Enhance the Effects of Anti-Inflammation Compared to Fibroblasts-Derived Exosomes. Vet Sci. 2021 Sep 3;8(9):182. doi: 10.3390/vetsci8090182. PMID: 34564576; PMCID: PMC8473240.

Author Response

Reviewer# 3

A manuscript (ID: vetsci-2933655) entitled: “Canine Mesenchymal Stem Cell Exosomes: State-of-the-art characterization, functional analysis and applications in various diseases has been submitted by the authors Mansur Abdullah Sandhu, Evelyn Saba and Alessandra Pelagalli to the journal Veterinary Sciences (Veterinary Physiology, Pharmacology, and Toxicology).

  1. The draft documents a solid piece of scientific work and should fit in a perfect way to the topics of the chosen target journal. However, I still have some remarks concerning the literature discussed. It is not quite clear to me why the authors have ignored an older publication regarding this topic. It would be of interest to learn a bit more of the progress in the field during the last ten years.

Evelien de Bakker, Bernadette Van Ryssen, Catharina De Schauwer & Evelyne Meyer (2013) Canine mesenchymal stem cells: state of the art, perspectives as therapy for dogs and as a model for man, Veterinary Quarterly, 33:4, 225-233, DOI: 10.1080/01652176.2013.873963

Author response: Respected reviewer, thank you for your comment, we tried to find as many relevant articles as we could, but some may have been overlooked. We have added this useful reference on page 2, (line 64-66) respectively.

  1. The authors focus on canine mesenchymal stem cells which are promising objects for the therapeutic approaches under study. Nevertheless, I have missed a detailed explanation what are the advantages of canine mesenchymal stem cells in comparison to stem cells from porcine or feline origin. 

Zhao, Q.; Zhang, X.; Li, Y.; He, Z.; Qin, K.; Buhl, E.M.; Mert, Ü.; Horst, K.; Hildebrand, F.; Balmayor, E.R.; et al. Porcine Mandibular Bone Marrow-Derived Mesenchymal Stem Cell (BMSC)-Derived Extracellular Vesicles Can Promote the Osteogenic Differentiation Capacity of Porcine Tibial-Derived BMSCs. Pharmaceutics 2024, 16, 279. https://doi.org/10.3390/pharmaceutics16020279

Sung SE, Seo MS, Kang KK, Choi JH, Lee S, Sung M, Kim K, Lee GW, Lim JH, Yang SY, Yim SG, Kim SK, Park S, Kwon YS, Yun S. Mesenchymal Stem Cell Exosomes Derived from Feline Adipose Tissue Enhance the Effects of Anti-Inflammation Compared to Fibroblasts-Derived Exosomes. Vet Sci. 2021 Sep 3;8(9):182. doi: 10.3390/vetsci8090182. PMID: 34564576; PMCID: PMC8473240.

Author response: Respected reviewer, we thank you for the time spent for the revision of our manuscript and for the appreciation of the topic that fit with the aim of the journal. We appreciate the precise comments regarding lack of an important literature reference on the perspective of therapy based on canine mesenchymal stem cells. We apologize for this oversight, which I think we missed while drafting the manuscript. As suggested by the reviewer, we added the previously mentioned references together with two more supporting references on page 2, lines 64-66 respectively.

Round 2

Reviewer 1 Report

Comments and Suggestions for Authors

All suggestions and criticisms presented were responded to appropriately.